# Convex Deep Learning via Normalized Kernels

**Özlem Aslan**
Dept of Computing Science
University of Alberta, Canada
ozlem@cs.ualberta.ca

**Xinhua Zhang**
Machine Learning Group
NICTA and ANU
xizhang@nicta.com.au

**Dale Schuurmans**
Dept of Computing Science
University of Alberta, Canada
dale@cs.ualberta.ca

## Abstract

Deep learning has been a long standing pursuit in machine learning, which until recently was hampered by unreliable training methods before the discovery of improved heuristics for embedded layer training. A complementary research strategy is to develop alternative modeling architectures that admit efficient training methods while expanding the range of representable structures toward deep models. In this paper, we develop a new architecture for nested nonlinearities that allows arbitrarily deep compositions to be trained to global optimality. The approach admits both parametric and nonparametric forms through the use of normalized kernels to represent each latent layer. The outcome is a fully convex formulation that is able to capture compositions of trainable nonlinear layers to arbitrary depth.

## 1  Introduction

Deep learning has recently achieved significant advances in several areas of perceptual computing, including speech recognition [1], image analysis and object detection [2, 3], and natural language processing [4]. The automated acquisition of representations is motivated by the observation that appropriate features make any learning problem easy, whereas poor features hamper learning. Given the practical significance of feature engineering, automated methods for feature discovery offer an important tool for applied machine learning. Ideally, automatically acquired features capture simple but salient aspects of the input distribution, upon which subsequent feature discovery can compose increasingly abstract and invariant aspects [5]; an intuition that appears to be well supported by recent empirical evidence [6].

Unfortunately, deep architectures are notoriously difficult to train and, until recently, required significant experience to manage appropriately [7, 8]. Beyond well known problems like local minima [9], deep training landscapes also exhibit plateaus [10] that arise from credit assignment problems in backpropagation. An intuitive understanding of the optimization landscape and careful initialization both appear to be essential aspects of obtaining successful training [11]. Nevertheless, the development of recent training heuristics has improved the quality of feature discovery at lower levels in deep architectures. These advances began with the idea of bottom-up, stage-wise unsupervised training of latent layers [12, 13] ("pre-training"), and progressed to more recent ideas like dropout [14]. Despite the resulting empirical success, however, such advances occur in the context of a problem that is known to be NP-hard in the worst case (even to approximate) [15], hence there is no guarantee that worst case versus "typical" behavior will not show up in any particular problem.

Given the recent success of deep learning, it is no surprise that there has been growing interest in gaining a deeper theoretical understanding. One key motivation of recent theoretical work has been to ground deep learning on a well understood computational foundation. For example, [16] demonstrates that polynomial time (high probability) identification of an optimal deep architecture can be achieved by restricting weights to bounded random variates and considering hard-threshold generative gates. Other recent work [17] considers a sum-product formulation [18], where guarantees can be made about the efficient recovery of an approximately optimal polynomial basis. Although these

treatments do not cover the specific models that have been responsible for state of the art results, they do provide insight into the computational structure of deep learning.

The focus of this paper is on kernel-based approaches to deep learning, which offer a potentially easier path to achieving a simple computational understanding. Kernels [19] have had a significant impact in machine learning, partly because they offer flexible modeling capability without sacrificing convexity in common training scenarios [20]. Given the convexity of the resulting training formulations, suboptimal local minima and plateaus are eliminated while reliable computational procedures are widely available. A common misconception about kernel methods is that they are inherently "shallow" [5], but depth is an aspect of how such methods are used and not an intrinsic property. For example, [21] demonstrates how nested compositions of kernels can be incorporated in a convex training formulation, which can be interpreted as learning over a (fixed) composition of hidden layers with infinite features. Other work has formulated adaptive learning of nested kernels, albeit by sacrificing convexity [22]. More recently, [23, 24] has considered learning kernel representations of latent clusters, achieving convex formulations under some relaxations. Finally, [25] demonstrated that an adaptive hidden layer could be expressed as the problem of learning a latent kernel between given input and output kernels within a jointly convex formulation. Although these works show clearly how latent kernel learning can be formulated, convex models have remained restricted to a single adaptive layer, with no clear paths suggested for a multi-layer extension.

In this paper, we develop a convex formulation of multi-layer learning that allows multiple latent kernels to be connected through nonlinear conditional losses. In particular, each pair of successive layers is connected by a prediction loss that is jointly convex in the adjacent kernels, while expressing a non-trivial, non-linear mapping between layers that supports multi-factor latent representations. The resulting formulation significantly extends previous convex models, which have only been able to train a single adaptive kernel while maintaining a convex training objective. Additional algorithmic development yields an approach with improved scaling properties over previous approaches, although not yet at the level of current deep learning methods. We believe the result is the first fully convex training formulation of a deep learning architecture with adaptive hidden layers, which demonstrates some useful potential in empirical investigations.

## 2 Background

To begin, consider a multi-layer conditional model where the input $\mathbf{x}_i$ is an $n$ dimensional feature vector and the output $\mathbf{y}_i \in \{0,1\}^m$ is a multi-label target vector over $m$ labels. For concreteness, consider a three-layer model (Figure 1). Here, the output of the first hidden layer

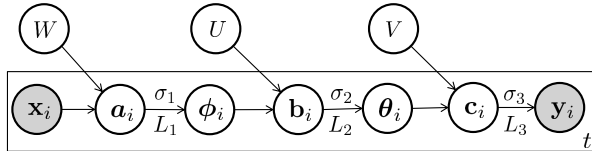

Figure 1: Multi-layer conditional models

is determined by multiplying the input, $\mathbf{x}_i$, with a weight matrix $W \in \mathbb{R}^{h \times n}$ and passing the result through a nonlinear transfer $\sigma_1$, yielding $\phi_i = \sigma_1(W\mathbf{x}_i)$. Then, the output of the second layer is determined by multiplying the first layer output, $\phi_i$, with a second weight matrix $U \in \mathbb{R}^{h' \times h}$ and passing the result through a nonlinear transfer $\sigma_2$, yielding $\theta_i = \sigma_2(U\phi_i)$, etc. The final output is then determined via $\hat{\mathbf{y}}_i = \sigma_3(V\theta_i)$, for $V \in \mathbb{R}^{m \times h'}$. For simplicity, we will set $h' = h$.

The goal of training is to find the weight matrices, $W$, $U$, and $V$, that minimize a training objective defined over the training data (with regularization). In particular, we assume the availability of $t$ training examples $\{(\mathbf{x}_i, \mathbf{y}_i)\}_{i=1}^t$, and denote the feature matrix $X := (\mathbf{x}_1, \ldots, \mathbf{x}_t) \in \mathbb{R}^{n \times t}$ and the label matrix $Y := (\mathbf{y}_1, \ldots, \mathbf{y}_t) \in \mathbb{R}^{m \times t}$ respectively. One of the key challenges for training arises from the fact that the latent variables $\Phi := (\phi_1, \ldots, \phi_t)$ and $\Theta := (\theta_1, \ldots, \theta_t)$ are unobserved.

To introduce our main development, we begin with a reconstruction of [25], which proposed a convex formulation of a simpler two-layer model. Although the techniques proposed in that work are intrinsically restricted to two layers, we will eventually show how this barrier can be surpassed through the introduction of a new tool—normalized output kernels. However, we first need to provide a more general treatment of the three main obstacles to obtaining a convex training formulation for multi-layer architectures like Figure 1.

### 2.1 First Obstacle: Nonlinear Transfers

The first key obstacle arises from the presence of the transfer functions, $\sigma_i$, which provide the essential nonlinearity of the model. In classical examples, such as auto-encoders and feed-forward neural

networks, an explicit form for $\sigma_i$ is prescribed, *e.g.* a step or sigmoid function. Unfortunately, the imposition of a nonlinear transfer in any deterministic model imposes highly *non-convex* constraints of the form: $\phi_i = \sigma_1(W\mathbf{x}_i)$. This problem is alleviated in nondeterministic models like probabilistic networks (PFN) [26] and restricted Boltzman machines (RBMs) [12], where the nonlinear relationship between the output (*e.g.* $\phi_i$) and the linear pre-image (*e.g.* $W\mathbf{x}_i$) is only softly enforced via a nonlinear *loss* $L$ that measures their discrepancy (see Figure 1). Such an approach was adopted by [25], where the values of the hidden layer responses (*e.g.* $\phi_i$) were treated as independent variables whose values are to be optimized in conjunction with the weights. In the present case, if one similarly optimizes rather than marginalizes over hidden layer values, $\Phi$ and $\Theta$ (*i.e.* Viterbi style training), a generalized training objective for a multi-layer architecture (Figure 1) can be expressed:

$$\min_{W,U,V,\Phi,\Theta} L_1(WX, \Phi) + \tfrac{1}{2}\|W\|^2 + L_2(U\Phi, \Theta) + \tfrac{1}{2}\|U\|^2 + L_3(V\Theta, Y) + \tfrac{1}{2}\|V\|^2 .^{[1]} \qquad (1)$$

The nonlinear loss $L_1$ bridges the nonlinearity introduced by $\sigma_1$, and $L_2$ bridges the nonlinearity introduced by $\sigma_2$, etc. Importantly, these losses, albeit nonlinear, can be chosen to be convex in their first argument; for example, as in standard models like PFNs and RBMs (implicitly). In addition to these exponential family models, which have traditionally been the focus of deep learning research, continuous latent variable models have also been considered, *e.g.* rectified linear model [27] and the exponential family harmonium. In this paper, like [25], we will use large-margin losses which offer additional sparsity and simplifications.

Unfortunately, even though the overall objective (1) is convex in the weight matrices $(W, U, V)$ *given* $(\Phi, \Theta)$, it is not *jointly* convex in all participating variables due to the interaction between the latent variables $(\Phi, \Theta)$ and the weight matrices $(W, U, V)$.

## 2.2 Second Obstacle: Bilinear Interaction

Therefore, the second key obstacle arises from the bilinear interaction between the latent variables and weight matrices in (1). To overcome this obstacle, consider a single connecting layer, which consists of an input matrix (*e.g.* $\Phi$) and output matrix (*e.g.* $\Theta$) and associated weight matrix (*e.g.* $U$):

$$\min_U L(U\Phi, \Theta) + \tfrac{1}{2}\|U\|^2 . \qquad (2)$$

By the representer theorem, it follows that the optimal $U$ can be expressed as $U = A\Phi'$ for some $A \in \mathbb{R}^{m \times t}$. Denote the linear response $Z = U\Phi = A\Phi'\Phi = AK$ where $K = \Phi'\Phi$ is the input kernel matrix. Then $\mathrm{tr}(UU') = \mathrm{tr}(AKA') = \mathrm{tr}(AKK^\dagger KA') = \mathrm{tr}(ZK^\dagger Z')$, where $K^\dagger$ is the Moore-Penrose pseudo-inverse (recall $KK^\dagger K = K$ and $K^\dagger KK^\dagger = K^\dagger$), therefore

$$(2) = \min_Z L(Z, \Theta) + \tfrac{1}{2}\mathrm{tr}(ZK^\dagger Z'). \qquad (3)$$

This is essentially the value regularization framework [28]. Importantly, the objective in (3) is *jointly convex* in $Z$ and $K$, since $\mathrm{tr}(ZK^\dagger Z)$ is a perspective function [29]. Therefore, although the single layer model is not jointly convex in the input features $\Phi$ and model parameters $U$, it is convex in the equivalent reparameterization $(K, Z)$ given $\Theta$. This is the technique used by [25] for the output layer. Finally note that $Z$ satisfies the constraint $Z \in \mathbb{R}^{m \times n}\Phi := \{U\Phi : U \in \mathbb{R}^{m \times n}\}$, which we will write as $Z \in \mathbb{R}\Phi$ for convenience. Clearly it is equivalent to $Z \in \mathbb{R}K$.

## 2.3 Third Obstacle: Joint Input-Output Optimization

The third key obstacle is that each of the latent variables, $\Phi$ and $\Theta$, simultaneously serve as the inputs and output targets for successive layers. Therefore, it is necessary to reformulate the connecting problem (2) so that it is jointly convex in all three components, $U$, $\Phi$ and $\Theta$; and unfortunately (3) is not convex in $\Theta$. Although this appears to be an insurmountable obstacle in general, [25] propose an exact reformulation in the case when $\Theta$ is boolean valued (consistent with the probabilistic assumptions underlying a PFM or RBM) by assuming the loss function satisfies an additional postulate.

**Postulate 1.** $L(Z, \Theta)$ *can be rewritten as* $L^u(\Theta'Z, \Theta'\Theta)$ *for* $L^u$ *jointly* convex in both arguments.

Intuitively, this assumption allows the loss to be parameterized in terms of the *propensity matrix* $\Theta'Z$ and the **u**nnormalized *output kernel* $\Theta'\Theta$ (hence the superscript of $L^u$). That is, the $(i, j)$-th component of $\Theta'Z$ stands for the linear response value of example $j$ with respect to the label of the example $i$. The $j$-th column therefore encodes the propensity of example $j$ to all other examples. This reparameterization is critical because it bypasses the linear response value, and relies solely on

the relationship between *pairs* of examples. The work [25] proposes a particular multi-label prediction loss that satisfies Postulate 1 for boolean target vectors $\boldsymbol{\theta}_i$; we propose an alternative below.

Using Postulate 1 and again letting $Z = U\Phi$, one can then rewrite the objective in (2) as $L^u(\Theta'U\Phi, \Theta'\Theta) + \frac{1}{2}\|U\|^2$. Now if we denote $N := \Theta'\Theta$ and $S := \Theta'Z = \Theta'U\Phi$ (hence $S \in \Theta'\mathbb{R}\Phi = N\mathbb{R}K$), the formulation can be reduced to the following (see Appendix A):

$$(2) = \min_S L^u(S, N) + \tfrac{1}{2}\operatorname{tr}(K^\dagger S' N^\dagger S). \tag{4}$$

Therefore, Postulate 1 allows (2) to be re-expressed in a form where the objective is jointly convex in the propensity matrix $S$ and output kernel $N$. Given that $N$ is a discrete but positive semidefinite matrix, a final relaxation is required to achieve a convex training problem.

**Postulate 2.** *The domain of $N = \Theta'\Theta$ can be relaxed to a convex set preserving sufficient structure.*

Below we will introduce an improved scheme for such relaxation. Although these developments support a convex formulation of two-layer model training [25], they appear insufficient for deeper models. For example, by applying (3) and (4) to the three-layer model of Figure 1, one obtains

$$L_1^u(S_1, N_1) + \tfrac{1}{2}\operatorname{tr}(K^\dagger S_1' N_1^\dagger S_1) + L_2^u(S_2, N_2) + \tfrac{1}{2}\operatorname{tr}(N_1^\dagger S_2' N_2^\dagger S_2) + L_3(Z_3, Y) + \tfrac{1}{2}\operatorname{tr}(Z_3 N_2^\dagger Z_3'),$$

where $N_1 = \Phi'\Phi$ and $N_2 = \Theta'\Theta$ are two latent kernels imposed between the input and output. Unfortunately, this objective is *not* jointly convex in all variables, since $\operatorname{tr}(N_1^\dagger S_2' N_2^\dagger S_2)$ is not jointly convex in $(N_1, S_2, N_2)$, hence the approach of [25] cannot extend beyond a single hidden layer.

## 3 Multi-layer Convex Modeling via Normalized Kernels

Although obtaining a convex formulation for general multi-layer models appears to be a significant challenge, progress can be made by considering an alternative approach. The failure of the previous development in [25] can be traced back to (2), which eventually causes the coupled, non-convex regularization to occur between connected latent kernels. A natural response therefore is to reconsider the original regularization scheme, keeping in mind that the representer theorem must still be supported. One such regularization scheme appears has been investigated in the clustering literature [30, 31], which suggests a reformulation of the connecting model (2) using value regularization [28]:

$$\min_U L(U\Phi, \Theta) + \tfrac{1}{2}\|\Theta'U\|^2. \tag{5}$$

Here $\|\Theta'U\|^2$ replaces $\|U\|^2$ from (2). The significance of this reformulation is that it still admits the representer theorem, which implies that the optimal $U$ must be of the form $U = (\Theta\Theta')^\dagger A\Phi'$ for some $A \in \mathbb{R}^{m \times n}$. Now, since $\Theta$ generally has full row rank (*i.e.* there are more examples than labels), one may execute a change of variables $A = \Theta B$. Such a substitution leads to the regularizer $\left\|\Theta'(\Theta\Theta')^\dagger \Theta B\Phi'\right\|^2$, which can be expressed in terms of the *normalized output kernel* [30]:

$$M := \Theta'(\Theta\Theta')^\dagger \Theta. \tag{6}$$

The term $(\Theta\Theta')^\dagger$ essentially normalizes the spectrum of the kernel $\Theta'\Theta$, and it is obvious that all eigen-values of $M$ are either 0 or 1, *i.e.* $M^2 = M$ [30]. The regularizer can be finally written as

$$\|MB\Phi'\|^2 = \operatorname{tr}(MBKB'M) = \operatorname{tr}(MBKK^\dagger KB'M) = \operatorname{tr}(SK^\dagger S'), \text{ where } S := MBK. \tag{7}$$

It is easy to show $S = \Theta'Z = \Theta'U\Phi$, which is exactly the propensity matrix.

As before, to achieve a convex training formulation, additional structure must be postulated on the loss function, but now allowing convenient expression in terms of normalized latent kernels.

**Postulate 3.** *The loss $L(Z, \Theta)$ can be written as $L^n(\Theta'Z, \Theta'(\Theta\Theta')^\dagger\Theta)$ where $L^n$ is* jointly *convex in both arguments. Here we write $L^n$ to emphasize the use of normalized kernels.*

Under Postulate 3, an alternative convex objective can be achieved for a local connecting model

$$L^n(S, M) + \tfrac{1}{2}\operatorname{tr}(SK^\dagger S'), \quad \text{where} \quad S \in M\mathbb{R}K. \tag{8}$$

Crucially, this objective is now jointly convex in $S$, $M$ and $K$; in comparison to (4), the normalization has removed the output kernel from the regularizer. The feasible region $\{(S, M, K) : M \succeq \mathbf{0}, K \succeq \mathbf{0}, S \in M\mathbb{R}K\}$ is also convex (see Appendix B). Applying (8) to the first two layers and (3) to the output layer, a fully convex objective for a multi-layer model (*e.g.*, as in Figure 1) is obtained:

$$L_1^n(S_1, M_1) + \tfrac{1}{2}\operatorname{tr}(S_1 K^\dagger S_1') + L_2^n(S_2, M_2) + \tfrac{1}{2}\operatorname{tr}(S_2 M_1^\dagger S_2') + L_3(Z_3, Y) + \tfrac{1}{2}\operatorname{tr}(Z_3 M_2^\dagger Z_3'), \tag{9}$$

where $S_1 \in M_1\mathbb{R}K$, $S_2 \in M_2\mathbb{R}M_1$, and $Z_3 \in \mathbb{R}M_2$.[2] All that remains is to design a convex relaxation of the domain of $M$ (for Postulate 2) and to design the loss $L^n$ (for Postulate 3).

### 3.1 Convex Relaxation of the Domain of Output Kernels $M$

Clearly from its definition (6), $M$ has a non-convex domain in general. Ideally one should design convex relaxations for each domain of $\Theta$. However, $M$ exhibits some nice properties for *any* $\Theta$:

$$M \succeq \mathbf{0}, \quad M \preceq I, \quad \operatorname{tr}(M) = \operatorname{tr}((\Theta\Theta')^{\dagger}(\Theta\Theta')) = \operatorname{rank}(\Theta\Theta') = \operatorname{rank}(\Theta). \tag{10}$$

Here $I$ is the identity matrix, and we also use $M \succeq \mathbf{0}$ to encode $M' = M$. Therefore, $\operatorname{tr}(M)$ provides a convenient proxy for controlling the rank of the latent representation, *i.e.* the number of hidden nodes in a layer. Given a specified number of hidden nodes $h$, we may enforce $\operatorname{tr}(M) = h$. The main relaxation introduced here is replacing the eigenvalue constraint $\lambda_i(M) \in \{0, 1\}$ (implied by $M^2 = M$) with $0 \le \lambda_i(M) \le 1$. Such a relaxation retains sufficient structure to allow, *e.g.*, a 2-approximation of optimal clustering to be preserved even by only imposing spectral constraints [30]. Experimental results below further demonstrate that nesting preserves sufficient structure, even with relaxation, to capture relationships that cannot be recovered by shallower architectures.

More refined constraints can be included to better account for the domain of $\Theta$. For example, if $\Theta$ expresses target values for a multiclass classification (*i.e.* $\Theta_{ij} \in \{0, 1\}$, $\Theta'\mathbf{1} = \mathbf{1}$ where $\mathbf{1}$ is a vector of all one's), we further have $M_{ij} \ge 0$ and $M\mathbf{1} = \mathbf{1}$. If $\Theta$ corresponds to multilabel classification where each example belongs to exactly $k$ (out of the $h$) labels (*i.e.* $\Theta \in \{0, 1\}^{h \times t}$, $\Theta'\mathbf{1} = k\mathbf{1}$), then $M$ can have negative elements, but the spectral constraint $M\mathbf{1} = \mathbf{1}$ still holds (see proof in Appendix C). So we will choose the domains for $M_1$ and $M_2$ in (9) to consist of the spectral constraints:

$$\mathcal{M} := \{\mathbf{0} \preceq M \preceq I : M\mathbf{1} = \mathbf{1}, \operatorname{tr}(M) = h\}. \tag{11}$$

### 3.2 A Jointly Convex Multi-label Loss for Normalized Kernels

An important challenge is to design an appropriate nonlinear loss to connect each layer of the model. Rather than conditional log-likelihood in a generative model, [25] introduced the idea of a using large margin, multi-label loss between a linear response, $\mathbf{z}$, and a boolean target vector, $\mathbf{y} \in \{0, 1\}^h$:

$$\tilde{L}(\mathbf{z}, \mathbf{y}) = \max(\mathbf{1} - \mathbf{y} + k\,\mathbf{z} - \mathbf{1}(\mathbf{y}'\mathbf{z})) \tag{12}$$

where $\mathbf{1}$ denotes the vector of all 1s. Intuitively this encourages the responses on the active labels, $\mathbf{y}'\mathbf{z}$, to exceed $k$ times the response of any inactive label, $kz_i$, by a margin, where the implicit nonlinear transfer is a step function. Remarkably, this loss can be shown to satisfy Postulate 1 [25].

This loss can be easily adapted to the normalized case as follows. We first generalize the notion of margin to consider a a "normalized label" $(YY')^{\dagger}\mathbf{y}$:

$$L(\mathbf{z}, \mathbf{y}) = \max(\mathbf{1} - (YY')^{\dagger}\mathbf{y} + k\,\mathbf{z} - \mathbf{1}(\mathbf{y}'\mathbf{z}))$$

To obtain some intuition, consider the multiclass case where $k = 1$. In this case, $YY'$ is a diagonal matrix whose $(i, i)$-th element is the number of examples in each class $i$. Dividing by this number allows the margin requirement to be weakened for popular labels, while more focus is shifted to less represented labels. For a given set of $t$ paired input/output pairs $(Z, Y)$ the sum of the losses can then be compactly expressed as $L(Z, Y) = \sum_j L(\mathbf{z}_j, \mathbf{y}_j) = \tau(kZ - (YY')^{\dagger}Y) + t - \operatorname{tr}(Y'Z)$, where $\tau(\Gamma) := \sum_j \max_i \Gamma_{ij}$. This loss can be shown to satisfy that satisfies Postulate 3:[3]

$$L^n(S, M) = \tau(S - \tfrac{1}{k}M) + t - \operatorname{tr}(S), \quad \text{where} \quad S = Y'Z \text{ and } M = Y'(YY')^{\dagger}Y. \tag{13}$$

This loss can be naturally interpreted using the remark following Postulate 1. It encourages that the propensity of example $j$ with respect to itself, $S_{jj}$, should be higher than its propensity with respect to other examples, $S_{ij}$, by a margin that is defined through the normalized kernel $M$. However note this loss does not correspond to a linear transfer between layers, even in terms of the propensity matrix $S$ or normalized output kernel $M$. As in all large margin methods, the initial loss (12) is a convex upper bound for an underlying discrete loss defined with respect to a step transfer.

## 4 Efficient Optimization

Efficient optimization for the multi-layer model (9) is challenging, largely due to the matrix pseudo-inverse. Fortunately, the constraints on $M$ are all spectral, which makes it easier to apply conditional gradient (CG) methods [32]. This is much more convenient than the models based on unnormalized kernels [25], where the presence of both spectral and non-spectral constraints necessitated expensive algorithms such as alternating direction method of multipliers [33].

**Algorithm 1:** Conditional gradient algorithm to optimize $f(M_1, M_2)$ for $M_1, M_2 \in \mathcal{M}$.

---

**1** Initialize $\tilde{M}_1$ and $\tilde{M}_2$ with some random matrices.

**2 while** $s = 1, 2, \ldots$ **do**

**3**     Compute the gradients $G_1 = \frac{\partial}{\partial M_1} f(\tilde{M}_1, \tilde{M}_2)$ and $G_2 = \frac{\partial}{\partial M_2} f(\tilde{M}_1, \tilde{M}_2)$.

**4**     Compute the new bases $M_1^s$ and $M_2^s$ by invoking oracle (15) with $G_1$ and $G_2$ respectively.

**5**     Totally corrective update: $\min_{\boldsymbol{\alpha} \in \Delta_s, \boldsymbol{\beta} \in \Delta_s} f\left(\sum_{i=1}^s \alpha_i M_1^i, \sum_{i=1}^s \beta_i M_2^i\right)$.

**6**     Set $\tilde{M}_1 = \sum_{i=1}^s \alpha_i M_1^i$ and $\tilde{M}_2 = \sum_{i=1}^s \beta_i M_2^i$; **break** if stopping criterion is met.

**7 return** $(\tilde{M}_1, \tilde{M}_2)$.

---

Denote the objective in (9) as $g(M_1, M_2, S_1, S_2, Z_3)$. The idea behind our approach is to optimize

$$f(M_1, M_2) := \min_{S_1 \in M_1 \mathbb{R} K, S_2 \in M_2 \mathbb{R} M_1, Z_3 \in \mathbb{R} M_2} g(M_1, M_2, S_1, S_2, Z_3) \tag{14}$$

by CG; see Algorithm 1 for details. We next demonstrate how each step can be executed efficiently.

**Oracle problem in Step 4.** This requires solving, given a gradient $G$ (which is real symmetric),

$$\max_{M \in \mathcal{M}} \operatorname{tr}(-GM) \Leftrightarrow \max_{0 \preceq M_1 \preceq I,\, \operatorname{tr}(M_1) = h-1} \operatorname{tr}(-G(HM_1 H + \tfrac{1}{t}\mathbf{1}\mathbf{1}')), \text{ where } H = I - \tfrac{1}{t}\mathbf{1}\mathbf{1}'. \tag{15}$$

Here we used Lemma 1 of [31]. By [34, Theorem 3.4], $\max_{0 \preceq M_1 \preceq I,\, \operatorname{tr}(M_1) = h-1} \operatorname{tr}(-HGHM_1) = \sum_{i=1}^{h-1} \lambda_i$ where $\lambda_1 \geq \lambda_2 \geq \ldots$ are the leading eigenvalues of $-HGH$. The maximum is attained at $M_1 = \sum_{i=1}^{h-1} \mathbf{v}_i \mathbf{v}_i'$, where $\mathbf{v}_i$ is the eigenvector corresponding to $\lambda_i$. The optimal solution to $\operatorname{argmax}_{M \in \mathcal{M}} \operatorname{tr}(-GM)$ can be recovered by $\sum_{i=1}^{h-1} \mathbf{v}_i \mathbf{v}_i' + \tfrac{1}{t}\mathbf{1}\mathbf{1}'$, which has low rank for small $h$.

**Totally corrective update in Step 5.** This is the most computationally intensive step of CG:

$$\min_{\boldsymbol{\alpha} \in \Delta_s,\, \boldsymbol{\beta} \in \Delta_s} f\left(\sum_{i=1}^s \alpha_i M_1^i, \sum_{i=1}^s \beta_i M_2^i\right), \tag{16}$$

where $\Delta_s$ stands for the $s$ dimensional probability simplex (sum up to 1). If one can solve (16) efficiently (which also provides the optimal $S_1, S_2, Z_3$ in (14) for the optimal $\boldsymbol{\alpha}$ and $\boldsymbol{\beta}$), then the gradient of $f$ can also be obtained easily by Danskin's theorem (for Step 3 of Algorithm 1). However, the totally corrective update is expensive because given $\boldsymbol{\alpha}$ and $\boldsymbol{\beta}$, each evaluation of the objective $f$ itself requires an optimization over $S_1, S_2$, and $Z_3$. Such a nested optimization can be prohibitive.

A key idea is to show that this totally corrective update can be accomplished with considerably improved efficiency through the use of block coordinate descent [35]. Taking into account the structure of the solution to the oracle, we denote

$$M_1(\boldsymbol{\alpha}) := \sum_i \alpha_i M_1^i = V_1 D(\boldsymbol{\alpha}) V_1', \quad \text{and} \quad M_2(\boldsymbol{\beta}) := \sum_i \beta_i M_2^i = V_2 D(\boldsymbol{\beta}) V_2', \tag{17}$$

where $D(\boldsymbol{\alpha}) = \operatorname{diag}([\alpha_1 \mathbf{1}_h', \alpha_2 \mathbf{1}_h', \ldots]')$ and $D(\boldsymbol{\beta}) = \operatorname{diag}([\beta_1 \mathbf{1}_h', \beta_2 \mathbf{1}_h', \ldots]')$. Denote

$$P(\boldsymbol{\alpha}, \boldsymbol{\beta}, S_1, S_2, Z_3) := g(M_1(\boldsymbol{\alpha}), M_2(\boldsymbol{\beta}), S_1, S_2, Z_3). \tag{18}$$

Clearly $S_1 \in M_1(\boldsymbol{\alpha}) \mathbb{R} K$ iff $S_1 = V_1 A_1 K$ for some $A_1$, $S_2 \in M_2(\boldsymbol{\beta}) \mathbb{R} M_1(\boldsymbol{\alpha})$ iff $S_2 = V_2 A_2 M_1(\boldsymbol{\alpha})$ for some $A_2$, and $Z_3 \in \mathbb{R} M_2(\boldsymbol{\beta})$ iff $Z_3 = A_3 M_2(\boldsymbol{\beta})$ for some $A_3$. So (16) is equivalent to

$$\min_{\boldsymbol{\alpha} \in \Delta_s,\, \boldsymbol{\beta} \in \Delta_s, A_1, A_2, A_3} P\left(\boldsymbol{\alpha}, \boldsymbol{\beta}, V_1 A_1 K, V_2 A_2 M_1(\boldsymbol{\alpha}), A_3 M_2(\boldsymbol{\beta})\right) \tag{19}$$

$$= L_1^n(V_1 A_1 K, M_1(\boldsymbol{\alpha})) + \tfrac{1}{2} \operatorname{tr}(V_1 A_1 K A_1' V_1') \tag{20}$$

$$+ L_2^n(V_2 A_2 M_1(\boldsymbol{\alpha}), M_2(\boldsymbol{\beta})) + \tfrac{1}{2} \operatorname{tr}(V_2 A_2 M_1(\boldsymbol{\alpha}) A_2' V_2') \tag{21}$$

$$+ L_3(A_3 M_2(\boldsymbol{\beta}), Y) + \tfrac{1}{2} \operatorname{tr}(A_3 M_2(\boldsymbol{\beta}) A_3'). \tag{22}$$

Thus we have eliminated all matrix pseudo-inverses. However, it is still expensive because the size of $A_i$ depends on $t$. To simplify further, assume $X'$, $V_1$ and $V_2$ all have full column rank.[4] Denote $B_1 = A_1 X'$ (note $K = X'X$), $B_2 = A_2 V_1$, $B_3 = A_3 V_2$. Noting (17), the objective becomes

$$R(\boldsymbol{\alpha}, \boldsymbol{\beta}, B_1, B_2, B_3) := L_1^n(V_1 B_1 X, V_1 D(\boldsymbol{\alpha}) V_1') + \tfrac{1}{2} \operatorname{tr}(V_1 B_1 B_1' V_1') \tag{23}$$

$$+ L_2^n(V_2 B_2 D(\boldsymbol{\alpha}) V_1', V_2 D(\boldsymbol{\beta}) V_2') + \tfrac{1}{2} \operatorname{tr}(V_2 B_2 D(\boldsymbol{\alpha}) B_2' V_2') \tag{24}$$

$$+ L_3(B_3 D(\boldsymbol{\beta}) V_2', Y) + \tfrac{1}{2} \operatorname{tr}(B_3 D(\boldsymbol{\beta}) B_3'). \tag{25}$$

This problem is much easier to solve, since the size of $B_i$ depends on the number of input features, the number of nodes in two latent layers, and the number of output labels. Due to the greedy nature of CG, the number of latent nodes is generally low. So we can optimize $R$ by block coordinate descent (BCD), *i.e.* alternating between:

1. Fix $(\boldsymbol{\alpha}, \boldsymbol{\beta})$, and solve $(B_1, B_2, B_3)$ (unconstrained smooth optimization, *e.g.* by LBFGS).
2. Fix $(B_1, B_2, B_3)$, and solve $(\boldsymbol{\alpha}, \boldsymbol{\beta})$ (*e.g.* by LBFGS with projection to simplex).

BCD is guaranteed to converge to a critical point when $L_1^n$, $L_2^n$ and $L_3$ are smooth.[5] In practice, these losses can be made smooth by, *e.g.* approximating the max in (13) by a softmax. It is crucial to note that although both of the two steps are convex, $R$ is *not* jointly convex in its variables. So in general, this alternating scheme can only produce a *stationary point* of $R$. Interestingly, we further show that *any* stationary point must provide a *global* optimal solution to $P$ in (18).

**Theorem 1.** *Suppose $(\boldsymbol{\alpha}, \boldsymbol{\beta}, B_1, B_2, B_3)$ is a stationary point of $R$ with $\alpha_i > 0$ and $\beta_i > 0$. Assume $X'$, $V_1$ and $V_2$ all have full column rank. Then it must be a* globally *optimal solution to $R$, and this $(\boldsymbol{\alpha}, \boldsymbol{\beta})$ must be an optimal solution to the totally corrective update* (16).

See the proof in Appendix D. It is noteworthy that the conditions $\alpha_i > 0$ and $\beta_i > 0$ are trivial to meet because CG is guaranteed to converge to optimal if $\alpha_i \geq 1/s$ and $\beta_i \geq 1/s$ at each step $s$.

## 5 Empirical Investigation

To investigate the potential of deep versus shallow convex training methods, and global versus local training methods, we implemented the approach outlined above for a three-layer model along with comparison methods. Below we use CVX3 and CVX2 to refer respectively to three and two-layer versions of the proposed model. For comparison, SVM1 refers to a one-layer SVM; and TS1a [37] and TS1b [38] refer to one-layer transductive SVMs; NET2 refers to a standard two-layer sigmoid neural network with hidden layer size chosen by cross-validation; and LOC3 refers to the proposed three-layer model with exact (unrelaxed) with local optimization. In these evaluations, we followed a similar transductive set up to that of [25]: a given set of data $(X, Y)$ is divided into separate training and test sets, $(X_L, Y_L)$ and $X_U$, where labels are only included for the training set. The training loss is then only computed on the training data, but the learned kernel matrices span the union of data. For testing, the kernel responses on test data are used to predict output labels.

### 5.1 Synthetic Experiments

Our first goal was to compare the effective modeling capacity of a three versus two-layer architecture given the convex formulations developed above. In particular, since the training formulation involves a convex relaxation of the normalized kernel domain, $\mathcal{M}$ in (11), it is important to determine whether the representational advantages of a three versus two-layer architecture are maintained. We conducted two sets of experiments designed to separate one-layer from two-layer or deeper models, and two-layer from three-layer or deeper models. Although separating two from one-layer models is straightforward, separating three from two-layer models is a subtler question. Here we considered two synthetic settings defined by basic functions over boolean features:

$$\text{Parity:} \quad y \ = \ x_1 \oplus x_2 \oplus \ldots \oplus x_n, \tag{26}$$

$$\text{Inner Product:} \quad y \ = \ (x_1 \wedge x_{m+1}) \oplus (x_2 \wedge x_{m+2}) \oplus \ldots \oplus (x_m \wedge x_n), \text{ where } m = \tfrac{n}{2}. \tag{27}$$

It is well known that Parity is easily computable by a two-layer linear-gate architecture but cannot be approximated by any one-layer linear-gate architecture on the same feature space [39]. The IP problem is motivated by a fundamental result in the circuit complexity literature: any small weights threshold circuit of depth 2 requires size $\exp(\Omega(n))$ to compute (27) [39, 40]. To generate data from

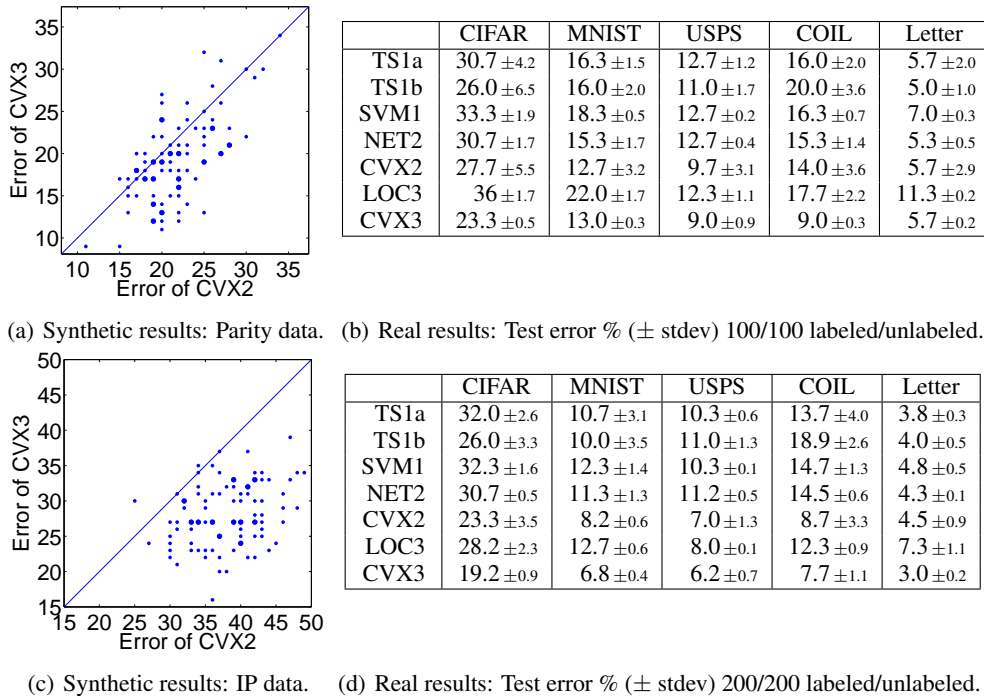

|  | CIFAR | MNIST | USPS | COIL | Letter |
|---|---|---|---|---|---|
| TS1a | $30.7_{\pm4.2}$ | $16.3_{\pm1.5}$ | $12.7_{\pm1.2}$ | $16.0_{\pm2.0}$ | $5.7_{\pm2.0}$ |
| TS1b | $26.0_{\pm6.5}$ | $16.0_{\pm2.0}$ | $11.0_{\pm1.7}$ | $20.0_{\pm3.6}$ | $5.0_{\pm1.0}$ |
| SVM1 | $33.3_{\pm1.9}$ | $18.3_{\pm0.5}$ | $12.7_{\pm0.2}$ | $16.3_{\pm0.7}$ | $7.0_{\pm0.3}$ |
| NET2 | $30.7_{\pm1.7}$ | $15.3_{\pm1.7}$ | $12.7_{\pm0.4}$ | $15.3_{\pm1.4}$ | $5.3_{\pm0.5}$ |
| CVX2 | $27.7_{\pm5.5}$ | $12.7_{\pm3.2}$ | $9.7_{\pm3.1}$ | $14.0_{\pm3.6}$ | $5.7_{\pm2.9}$ |
| LOC3 | $36_{\pm1.7}$ | $22.0_{\pm1.7}$ | $12.3_{\pm1.1}$ | $17.7_{\pm2.2}$ | $11.3_{\pm0.2}$ |
| CVX3 | $23.3_{\pm0.5}$ | $13.0_{\pm0.3}$ | $9.0_{\pm0.9}$ | $9.0_{\pm0.3}$ | $5.7_{\pm0.2}$ |

(a) Synthetic results: Parity data. (b) Real results: Test error % ($\pm$ stdev) 100/100 labeled/unlabeled.

|  | CIFAR | MNIST | USPS | COIL | Letter |
|---|---|---|---|---|---|
| TS1a | $32.0_{\pm2.6}$ | $10.7_{\pm3.1}$ | $10.3_{\pm0.6}$ | $13.7_{\pm4.0}$ | $3.8_{\pm0.3}$ |
| TS1b | $26.0_{\pm3.3}$ | $10.0_{\pm3.5}$ | $11.0_{\pm1.3}$ | $18.9_{\pm2.6}$ | $4.0_{\pm0.5}$ |
| SVM1 | $32.3_{\pm1.6}$ | $12.3_{\pm1.4}$ | $10.3_{\pm0.1}$ | $14.7_{\pm1.3}$ | $4.8_{\pm0.5}$ |
| NET2 | $30.7_{\pm0.5}$ | $11.3_{\pm1.3}$ | $11.2_{\pm0.5}$ | $14.5_{\pm0.6}$ | $4.3_{\pm0.1}$ |
| CVX2 | $23.3_{\pm3.5}$ | $8.2_{\pm0.6}$ | $7.0_{\pm1.3}$ | $8.7_{\pm3.3}$ | $4.5_{\pm0.9}$ |
| LOC3 | $28.2_{\pm2.3}$ | $12.7_{\pm0.6}$ | $8.0_{\pm0.1}$ | $12.3_{\pm0.9}$ | $7.3_{\pm1.1}$ |
| CVX3 | $19.2_{\pm0.9}$ | $6.8_{\pm0.4}$ | $6.2_{\pm0.7}$ | $7.7_{\pm1.1}$ | $3.0_{\pm0.2}$ |

(c) Synthetic results: IP data. (d) Real results: Test error % ($\pm$ stdev) 200/200 labeled/unlabeled.

Figure 2: Experimental results (synthetic data: larger dots mean repetitions fall on the same point).

these models, we set the number of input features to $n = 8$ (instead of $n = 2$ as in [25]), then generate 200 examples for training and 100 examples for testing; for each example, the features $x_i$ were drawn from $\{0, 1\}$ with equal probability. Then each $x_i$ was corrupted independently by a Gaussian noise with zero mean and variance 0.3. The experiments were repeated 100 times, and the resulting test errors of the two models are plotted in Figure 2. Figure 2(c) clearly shows that CVX3 is able to capture the structure of the IP problem much more effectively than CVX2, as the theory suggests for such architectures. In almost every repetition, CVX3 yields a lower (often much lower) test error than CVX2. Even on the Parity problem (Figure 2(a)), CVX3 generally produces lower error, although its advantage is not as significant. This is also consistent with theoretical analysis [39, 40], which shows that IP is harder to model than parity.

## 5.2 Experiments on Real Data

We also conducted an empirical investigation on some real data sets. Here we tried to replicate the results of [25] on similar data sets, USPS and COIL from [41], Letter from [42], MNIST, and CIFAR-100 from [43]. Similar to [23], we performed an optimistic model selection for each method on an initial sample of $t$ training and $t$ test examples; then with the parameters fixed the experiments were repeated 5 times on independently drawn sets of $t$ training and $t$ test examples from the remaining data. The results shown in Table 2(b) and Table 2(d) show that CVX3 is able to systematically reduce the test error of CVX2. This suggests that the advantage of deeper modeling does indeed arise from enhanced representation ability, and not merely from an enhanced ability to escape local minima or walk plateaus, since neither exist in these cases.

## 6 Conclusion

We have presented a new formulation of multi-layer training that can accommodate an arbitrary number of nonlinear layers while maintaining a jointly convex training objective. Accurate learning of additional layers, when required, appears to demonstrate a marked advantage over shallower architectures, even when models can be trained to optimality. Aside from further improvements in algorithmic efficiency, an interesting direction for future investigation is to capture unsupervised "stage-wise" training principles via auxiliary autoencoder objectives within a convex framework, rather than treating input reconstruction as a mere heuristic training device.

## Footnotes

[1] The terms $\|W\|^2$, $\|U\|^2$ and $\|V\|^2$ are regularizers, where the norm is the Frobenius norm. For clarity we have omitted the regularization parameters, relative weightings between different layers, and offset weights from the model. These components are obviously important in practice, however they play no key role in the technical development and removing them greatly simplifies the expressions.

[2] Clearly the first layer can still use (4) with an unnormalized output kernel $N_1$ since its input $X$ is observed.

[3] A simple derivation extends [25]: $\tau(kZ - (YY')^{\dagger}Y) = \max_{\Lambda:\mathbb{R}_+^{m \times t}:\Lambda'\mathbf{1}=\mathbf{1}} \operatorname{tr}(\Lambda'(kZ - (YY')^{\dagger}Y)) = \max_{\Omega:\mathbb{R}_+^{t \times t}:\Omega'\mathbf{1}=\mathbf{1}} \frac{1}{k}\operatorname{tr}(\Omega'Y'(kZ - (YY')^{\dagger}Y)) = \tau(Y'Z - \frac{1}{k}M)$. Here the second equality follows because for any $\Lambda \in \mathbb{R}_+^{m \times t}$ satisfying $\Lambda'\mathbf{1} = \mathbf{1}$, there must be an $\Omega \in \mathbb{R}_+^{t \times t}$ satisfying $\Omega'\mathbf{1} = \mathbf{1}$ and $\Lambda = Y\Omega/k$.

[4] This assumption is valid provided the features in $X$ are linearly independent, since the bases (eigenvectors) accumulated through all iterations so far are also independent. The only exception is the eigen-vector $\frac{1}{\sqrt{t}}\mathbf{1}$. But since $\boldsymbol{\alpha}$ and $\boldsymbol{\beta}$ lie on a simplex, it always contributes a *constant* $\frac{1}{t}\mathbf{1}\mathbf{1}'$ to $M_1(\boldsymbol{\alpha})$ and $M_2(\boldsymbol{\beta})$.

[5]Technically, for BCD to converge to a critical point, each block optimization needs to have a unique optimal solution. To ensure uniqueness, we used a method equivalent to the proximal method in Proposition 7 of [36].

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
