[Supplementary Material · Appendix.pdf]

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

# Supplementary material

## A   Derivation of (4)

Recall that the objective function in (2) can be rewritten as

$$L^u(\Theta' U\Phi, \Theta'\Theta) + \frac{1}{2}\|U\|^2, \tag{28}$$

using Postulate 1. Then by the representer theorem, the optimal $U$ can be expressed by $U = \Theta A\Phi'$ for some matrix $A$. Denoting $S := \Theta'Z = \Theta'U\Phi$ and $N := \Theta'\Theta$, the objective becomes

$$L^u(S, N) + \frac{1}{2}\operatorname{tr}(K^\dagger S' N^\dagger S), \tag{29}$$

since

$$\operatorname{tr}(K^\dagger S' N^\dagger S) = \operatorname{tr}(K^\dagger \Phi' U' \Theta N^\dagger \Theta' U\Phi) = \operatorname{tr}(K^\dagger \Phi' \Phi A' \Theta' \Theta N^\dagger \Theta' \Theta A\Phi'\Phi) \tag{30}$$

$$= \operatorname{tr}(K^\dagger K A' N N^\dagger N A K) = \operatorname{tr}(K A' N A) \quad (\text{using } KK^\dagger K = K) \tag{31}$$

$$= \operatorname{tr}(\Phi'\Phi A'\Theta'\Theta A) = \|\Theta A\Phi'\|^2 = \|U\|^2. \tag{32}$$

## B   Proof that the Feasible Region of (8) is Convex

Here we prove that the set

$$V := \{(M, S, K) : M \succeq \mathbf{0}, K \succeq \mathbf{0}, S \in M\mathbb{R}K\} \tag{33}$$

is convex.

Recall that $\mathbb{R}\Phi$ consists of matrices in the row span of $\Phi$. Since $M$ and $K$ are PSD, the constraint $S \in M\mathbb{R}K$ involves the intersection of two convex constraints. Let

$$V_1 = \{(M, S, K) : M \succeq \mathbf{0}, K \succeq \mathbf{0}, S \in M\mathbb{R}\} \tag{34}$$

$$V_2 = \{(M, S, K) : M \succeq \mathbf{0}, K \succeq \mathbf{0}, S \in \mathbb{R}K\}. \tag{35}$$

First, we show that the set $V_1$ is convex. Suppose $S_1 \in M_1\mathbb{R}$ and $S_2 \in M_2\mathbb{R}$ where $M_1 \succeq \mathbf{0}$ and $M_2 \succeq \mathbf{0}$. To prove that $S_1 + S_2 \in (M_1 + M_2)\mathbb{R}$, it suffices to show that $M_1\mathbb{R} \cup M_2\mathbb{R} \subseteq (M_1 + M_2)\mathbb{R}$. To this end, consider its contrapositive, $i.e.$, there exists $\mathbf{r}_1$, $\mathbf{r}_2$, $\mathbf{x}$ such that $\mathbf{x}'(M_1\mathbf{r}_1 + M_2\mathbf{r}_2) \neq 0$ while $\mathbf{x}'(M_1 + M_2) = \mathbf{0}$. However this is impossible, because the latter implies $\mathbf{x}'M_1 = \mathbf{x}'M_2 = \mathbf{0}'$ when $M_1 \succeq \mathbf{0}$ and $M_2 \succeq \mathbf{0}$.

$V_2$ is convex for isomorphic reasons. It remains only to show $V = V_1 \cap V_2$, since an intersection of convex sets is convex. The $\subseteq$ relationship is straightforward. To show $\supseteq$, let $S = MP = QK$ for some $P$ and $Q$; then note that $SK^\dagger K = S = MM^\dagger S$. Hence $MM^\dagger SK^\dagger K = S$, $i.e.$ $S \in V$.

## C   Proof of $M\mathbf{1} = \mathbf{1}$ in Section 3.1

Proof: Consider the compact SVD: $\Theta = U\Sigma V'$ where $U'U = I$ and $V'V = I$. Then $\Theta'\mathbf{1} = k\mathbf{1}$ implies $\Sigma U'\mathbf{1} = kV'\mathbf{1}$. Since $M = VV'$: $M\mathbf{1} = VV'\mathbf{1} = \frac{1}{k}V\Sigma U'\mathbf{1} = \frac{1}{k}\Theta'\mathbf{1} = \mathbf{1}$. Note $\Theta_{ij} \in \{0, 1\}$ is not used.

## D   Proof of Theorem 1

For convenient reference, we repeat the objective $R$ here

$$R(\boldsymbol{\alpha}, \boldsymbol{\beta}, B_1, B_2, B_3) := L_1^n(V_1 B_1 X, V_1 D(\boldsymbol{\alpha})V_1') + \frac{1}{2}\operatorname{tr}(V_1 B_1 B_1' V_1') \tag{36}$$

$$+ L_2^n(V_2 B_2 D(\boldsymbol{\alpha})V_1', V_2 D(\boldsymbol{\beta})V_2') + \frac{1}{2}\operatorname{tr}(V_2 B_2 D(\boldsymbol{\alpha})B_2' V_2') \tag{37}$$

$$+ L_3(B_3 D(\boldsymbol{\beta})V_2', Y) + \frac{1}{2}\operatorname{tr}(B_3 D(\boldsymbol{\beta})B_3'). \tag{38}$$

To enforce Theorem 1, we will use constraints

$$\alpha_i \geq 1/s, \quad \beta_i \geq 1/s, \quad \mathbf{1}'\boldsymbol{\alpha} = 1, \quad \mathbf{1}'\boldsymbol{\beta} = 1. \tag{39}$$

*Proof.* Let

$$C_2 = B_2 D(\boldsymbol{\alpha}), \qquad C_3 = B_3 D(\boldsymbol{\beta}) \tag{40}$$

Then $R$ is equivalent to

$$R(\boldsymbol{\alpha}, \boldsymbol{\beta}, B_1, B_2, B_3) = S(\boldsymbol{\alpha}, \boldsymbol{\beta}, B_1, C_2, C_3) \tag{41}$$

$$:= L_1^n(V_1 B_1 X, V_1 D(\boldsymbol{\alpha})V_1') + \frac{1}{2}\operatorname{tr}(V_1 B_1 B_1' V_1') \tag{42}$$

$$+ L_2^n(V_2 C_2 V_1', V_2 D(\boldsymbol{\beta})V_2') + \frac{1}{2}\operatorname{tr}(V_2 C_2 D(\boldsymbol{\alpha})^\dagger C_2' V_2) \tag{43}$$

$$+ L_3(C_3 V_2', Y) + \frac{1}{2}\operatorname{tr}(C_3 D(\boldsymbol{\beta})^\dagger C_3'). \tag{44}$$

Clearly $S$ is jointly convex in $(\boldsymbol{\alpha}, \boldsymbol{\beta}, B_1, C_2, C_3)$. Thanks to the invertability of $D(\boldsymbol{\alpha})$ and $D(\boldsymbol{\beta})$, we apply chain rule to the stationarity condition of $B_i$:

$$\mathbf{0} = \frac{\partial}{\partial B_1}R(\boldsymbol{\alpha}, \boldsymbol{\beta}, B_1, B_2, B_3) = \frac{\partial}{\partial B_1}S(\boldsymbol{\alpha}, \boldsymbol{\beta}, B_1, C_2, C_3) \Rightarrow \frac{\partial}{\partial B_1}S(\boldsymbol{\alpha}, \boldsymbol{\beta}, B_1, C_2, C_3) = \mathbf{0} \tag{45}$$

$$\mathbf{0} = \frac{\partial}{\partial B_2}R(\boldsymbol{\alpha}, \boldsymbol{\beta}, B_1, B_2, B_3) = \frac{\partial}{\partial C_2}S(\boldsymbol{\alpha}, \boldsymbol{\beta}, B_1, C_2, C_3)D(\boldsymbol{\alpha}) \Rightarrow \frac{\partial}{\partial C_2}S(\boldsymbol{\alpha}, \boldsymbol{\beta}, B_1, C_2, C_3) = \mathbf{0} \tag{46}$$

$$\mathbf{0} = \frac{\partial}{\partial B_3}R(\boldsymbol{\alpha}, \boldsymbol{\beta}, B_1, B_2, B_3) = \frac{\partial}{\partial C_3}S(\boldsymbol{\alpha}, \boldsymbol{\beta}, B_1, C_2, C_3)D(\boldsymbol{\beta}) \Rightarrow \frac{\partial}{\partial C_3}S(\boldsymbol{\alpha}, \boldsymbol{\beta}, B_1, C_2, C_3) = \mathbf{0} \tag{47}$$

Note $R$ is convex in $(\boldsymbol{\alpha}, \boldsymbol{\beta})$ given $B_i$, and $\alpha_i, \beta_i$ satisfy the constraint (39). So KKT conditions (regarding the optimality of $(\boldsymbol{\alpha}, \boldsymbol{\beta})$ given $B_1, B_2, B_3$) ensure that there exist Lagrange multipliers $(\mu, \lambda)$ (corresponding to the sum up to one constraints) and $(a_i, b_i)$ (corresponding to the greater than or equal to $1/s$ constraint) such that

$$\alpha_i \geq \frac{1}{s}, \quad \beta_i \geq \frac{1}{s}, \quad \mathbf{1}'\boldsymbol{\alpha} = 1, \quad \mathbf{1}'\boldsymbol{\beta} = 1, \tag{48}$$

$$a_i(\alpha_i - \tfrac{1}{s}) = 0, \quad b_i(\beta_i - \tfrac{1}{s}) = 0, \quad a_i \geq 0, \quad b_i \geq 0, \tag{49}$$

and

$$0 = \frac{\partial}{\partial \alpha_i}R(\boldsymbol{\alpha}, \boldsymbol{\beta}, B_1, B_2, B_3) - \mu - a_i \tag{50}$$

$$\text{(chain rule)} = \frac{\partial}{\partial \alpha_i}S(\boldsymbol{\alpha}, \boldsymbol{\beta}, B_1, C_2, C_3) + \left\langle \frac{\partial}{\partial \alpha_i}B_2 D(\boldsymbol{\alpha}), \frac{\partial}{\partial C_2}S(\boldsymbol{\alpha}, \boldsymbol{\beta}, B_1, C_2, C_3) \right\rangle - \mu - a_i \tag{51}$$

$$\text{by (46)} = \frac{\partial}{\partial \alpha_i}S(\boldsymbol{\alpha}, \boldsymbol{\beta}, B_1, C_2, C_3) - \mu - a_i, \tag{52}$$

and similarly,

$$0 = \frac{\partial}{\partial \beta_i}S(\boldsymbol{\alpha}, \boldsymbol{\beta}, B_1, C_2, C_3) - \lambda - b_i. \tag{53}$$

Since $S$ is jointly convex in all its variables, (45)-(49), (52), and (53) provide all the KKT conditions required to establish the global optimality of $(\boldsymbol{\alpha}, \boldsymbol{\beta}, B_1, C_2, C_3)$ for $S$. The claims in the theorem then follow. $\qquad\square$