[Reviews · NeurIPS 2014]

Submitted by Assigned_Reviewer_27

The submission describes a convex deep learning formulation that leverages a number of key ideas. First, a training objective is proposed that explicitly includes the outputs of hidden layers as variables to be inferred via optimization. These are linked to linear responses via a loss function, and the net objective is the sum of these loss functions across the layers, plus some regularization terms. Next, a number of changes of variables are performed in order to reparameterize the objective into a convex form, heavily leveraging the representer theorem and the idea of value regularization. We are left with a convex objective in terms of three different matrices (per layer) to optimize. In particular, one of these matrices is a nonparametric ‘normalized output kernel’ matrix, which takes the place of optimizing over the hidden layer outputs directly; however, this leads to a transductive method where we must simultaneously solve the optimization for training and test inputs. Finally, a relaxation must be performed in order to obtain a truly convex formulation, since the set of valid kernel matrices is generally non convex (possibly discrete, given assumptions on the hidden layer values). Experiments are shown comparing the method to a previous two-layer convex model on small datasets.

Quality:
The basic ideas presented seem technically sound, although I was unable to rigorously verify many aspects of the derivation. It is no small technical feat to achieve a convex deep learning formulation, and the authors make use of several very sophisticated tricks, gyrations, assumptions, and clever observations to achieve this goal. Most of the derivation of the method consists of reparameterizations of the objective that leverage the representer theorem.

However impressive this technical feat may be, my overall impression is that the resulting method is more of an academic curiosity than a practical method, for a few reasons. First, the derivation of the method makes a few critical assumptions and relaxations to achieve a convex objective, such that we are left wondering whether the resulting model really captures the essence of a traditional deep network. The most significant of these is relaxing the set of output kernels to a convex set. The original motivation for the objective seemed to be a Boltzmann-machine-style model, where the hidden units are restricted to take on binary values--except the submission proposes to infer the hidden states deterministically. The convex relaxation of the output kernel then effectively relaxes this binary constraint on the hidden units, which brings us a little farther away from the original motivation, and raises doubts as to how the new objective might be interpreted. A similar argument can be made for the loss, which seems to replace a saturating loss with a hinge loss. The submission gives little intuition as to why this is justified. The transductive nature of the algorithm is also disappointing, and doesn’t seem scalable to large datasets.

Unfortunately, I found several important steps in the derivation to be beyond my ability to comprehend. However, I have at least one technical doubt that the authors may want to address—the reparameterizations of the objective induce constraints on the matrices that are not obviously convex constraints: for example, S \in MRK in Eq. 8, where M and K are in convex sets, and R is an appropriate matrix. It is not clear to me whether this set is convex. Perhaps the authors could provide a reference. Also, I think \Phi U in Eq. 5 should be U \Phi.

The experiments are very weak and add to my doubts regarding the practicality of the method. The datasets used are small subsets of the original sets, leading me to believe that scalability is indeed an issue. The only comparison is to a two-layer version of the method. An obvious missing comparison is to a traditional deep net. For that matter, even a comparison to a shallow method should at least be included as a baseline. Also, I would be interested to see how nonlinear optimization of the original non convex objective fares compared to the convex version.

Clarity:
A lack of clarity is perhaps the weakest point of the submission. The authors seem to have in mind a particular mental model and rationalization for their approach that does not come through in the submission. For example, in several places, passing remarks are made about the nature of the loss function (e.g., convexity in one slot or the other), but no such assumptions are clearly made up-front. It would also very much help to clearly delineate the connection and inspiration from neural-net-like Bayes nets or Boltzmann machines in order better motivate the objective. For example, what might the loss look like concretely for a Bayes net, and how does this compare to the actual loss used? What effects does the difference have on the expressive capacity of the method? The submission should makes these issues more clear.

I also found several critical points in the derivation to be lacking any kind of meaningful explanation. This is especially true in the vicinity of Eq. 5-7. This is the core of the method, but very little explanation is provided besides a few references to value regularization and some recent papers on clustering. This is very disappointing, because I found section 2 to be fairly readable in comparison, but that section essentially ends in a dead-end. I would have preferred it if section 2 were much shorter, and section 3 were longer.

Originality:
Although this work seems to be inspired by earlier work on two-level convex networks, and is based on the same basic model, the trick to generalize to arbitrary nesting seems novel enough. The proposed optimization algorithm also seems novel.

Significance:
As the paper itself acknowledges, obtaining a convex training formulation is not a universally accepted aim of deep learning researchers, nor do local minima seem to be a serious problem in real networks. The proposed training procedure is fairly complicated and seems to scale poorly, judging by the experiments. The formulation is fairly inflexible, working only with a narrowly defined set of loss functions, and the resulting algorithm is transductive, which further hurts performance and ultimately limits scalability. In the end, even though the optimization is convex, this is only obtained by making a critical relaxation that limits our ability to constrain the domain of the hidden layer outputs. For all these reasons, the immediate practical impact of this work is likely to be very low.

On the other hand, it is possible that this work could stimulate further work on the subject, since the very idea that a convex formulation might capture the important aspects of a deep network is interesting by itself. Future work might elucidate the intrinsic theoretical limitations of this formulation versus traditional deep nets.
Summary: The main result that a deep architecture can be expressed in a convex objective is interesting; however, little insight is given as to the limitations in expressivity of the method compared to standard (non convex) formulations, and the method does not seem practical at all at this point.

Submitted by Assigned_Reviewer_39

The authors tackle here the difficult for proposing convex relaxations for learning deep architectures. The main difficulty in learning deep architectures lies in the fact that the model is built by iteratively performing composition of layers. For each layer, the key quantities are the weight matrix (parameterizing the inputs) and the output matrix (coding the output labels). An output matrix of one layer enters the input quantities of the next layer, which is the crux of the difficulty of posing the learning problem for learning deep architectures.

The authors propose here a new convex relaxation approach based on a previous relaxation that was made popular for maximum margin clustering. For a given single-layer learning problem such as maximum margin clustering, the convex relaxation used in maximum margin clustering consists in lifting the problem in a higher-dimensional space by parametrizing it using the equivalence matrix (Theta Theta') instead of the output matrix (Theta). In this paper, the authors propose instead to base the convex relaxation on the *normalized* quantity M = Theta' (Theta Theta')# Theta, where # denotes the pseudo-inverse and ' the transpose. This new convex relaxation has significant interesting outcomes. In particular, with appropriate modelling, the proposed approach based on this relaxation leads to constraints on M that are all spectral, whereas the un-normalized convex relaxation would lead to both spectral constraints and other non-spectral constraints. Owing to this elegant formulation, the approach can be efficiently implemented with a conditional gradient algorithm. Preliminary experimental results are presented on synthetic and real-world datasets.

Detailed comments

The argument (lines 349-352) about the convergence to a stationary point is rather terse in the current version of the paper and should be significantly clarified in the final version of the paper. The exposition could be improved, by making extensive use of the supplemental material for the proofs and mathematical arguments and using the main part of the paper for clear explanations and illustrations.

I would like to praise the (scholar) effort of the authors to carefully define what the problem is, for designing a useful convex relaxation, by reviewing many possible options and highlighting their corresponding challenges. Putting aside the main contributions of the paper, these sections of the paper already represent a very valuable review material in order to tackle the current challenges of building theoretically-grounded deep learning approaches. I seldom read papers with such a significant and unusual effort. I'm actually rather impressed by such an effort for a conference paper.
Summary: The paper is fresh and innovative. Designing useful convex relaxations of deep learning problems is probably one of the most daunting challenge in the ML community right now. This paper proposes a novel and interesting approach. A clear accept.

Submitted by Assigned_Reviewer_41

This paper studies a a multi-layer conditional model for deep learning, where the input is sequentially multiplied by some weight matrices and then passed through nonlinear transfer functions. The aim is to choose the weight matrices so as to match the known multi-label target vectors of the training data and to keep some norms of the weight matrices small (regularization). Each non-terminal layer of the model produces unobserved outputs, which are treated as latent variables. If there are only two layers, it has been shown in reference [26] that the optimal weight matrices can be found approximately by solving a tractable convex optimization problem. This convex formulation is facilitated by the following tricks:

(1) The nonlinear transfer constraints linking the inputs and outputs of each layer are softly enforced by a nonlinear penalty function, which can be chosen to be linear in the layer's weight matrix.
(2) The bilinear coupling between the weight matrices and the latent variables is eliminated by a coordinate transformation whereby the problem is reformulated in terms of each layer's linear response and input kernel matrix.

If there is a third layer, one needs to find a (approximate) reformulation that is jointly convex in the inputs, latent variables and outputs of any intermediate layer. If the latent variables are Boolean-valued, [26] propose an exact reformulation under the additional assumptions that the penalty functions are convex in the propensity matrix and the output kernel and that the domain of the output kernel admits a tight convex relaxation.

This paper builds on the ideas from [26] to develop an approximate convex formulation for general models with intermediate layers. The key enabling mechanism is to introduce normalized kernels, which are defined in a similar way as the ordinary kernels but are normalized such that all of their eigenvalues are either 0 and 1. Moreover, the classical regularization is replaced by value regularization (penalizing the norm of the weight matrices multiplied with the outputs of the layer), and the penalty functions encoding the nonlinear relationship between weighted inputs and outputs are assumed to adopt a specific form (where the implicit linear transfer functions are step functions). Finally, a tight convex relaxation of the domain of the normalized output kernels is required (it is suggested to use simple spectral constraints).

The paper also suggests a nested optimization algorithm to optimize the training objective. The outer optimization is based on a conditional gradient algorithm and optimizes the best training objective as a function of the normalized kernels (i.e., essentially setting all propensities to their optimal values given the kernels). This algorithm involves a totally corrective update step, which is carried out efficiently via block coordinate descent.

The idea to use normalized kernels is a clever one that clearly deserves to be published. The numerical tests show that three-layer models can offer tangible advantages over two-layer architectures both in synthetic experiments as well as in experiments with real data.

It is shown that the loss function of the final formulation is jointly convex in all optimization variables (normalized kernels and propensities). However, the problem still contains non-standard constraints which required the propensities to be representable as matrix products involving the input and output kernels of the respective layer. These constraints are per se nonconvex. So unless I am missing something, this paper does not offer a fully convex model for training multi-layer models. If my understanding is correct, I feel that the authors overstate their contribution. Generally, it would be good to explicitly list all approximations that are made when passing from the desired model to the one that is actually solved.

I have the impression that the exposition could be improved by writing down the full 2 layer model from [26] and the full three- (or multi)-level model proposed here (including all constraints). In order to develop a better understanding of the new formulation with normalized kernels, it would also have been nice to discuss the difference between this approach and the approach from [26] when both are applied to a two-layer model. What are the differences between the two approaches in this special case?

It was also not clear to me whether the methods from [26] extend to multi-layer architectures whenever all latent variables are Boolean-valued. This point should be clarified.

On l352 it is stated that R is not jointly convex in its variables and that therefore an alternating optimization scheme can only produce a stationary point of R. In general, however, I disagree that the limit is a stationary point/local minimum. So the statement that a stationary point of R can be found may already be too optimistic. Here is an example of a convex maximization problem with linear constraints and a concave objective function for which an alternating optimization scheme gets stuck at a non-stationary point: max f(x,y) s.t. 0<=x<=1, 0<=y<=1, where f(x,y)= min{-x+2*y, 2*x-y} is jointly concave in x and y. If we start at point (1,0) and optimize w.r.t. x (fixing y=0), we will end up in point (0,0). Optimizing over y will not get us away from this point. That is, we are stuck at (0,0), which is not a local optimum of f. It is not clear to me why the proposed block coordinate descent algorithm described in lines 348/349 should not suffer from similar problems. If we are not certain to find a stationary point of R, then the relevance of Theorem 1, one of the central contributions of the paper, remains unclear.

Specific comments:

l212: Say explicitly that the squared Frobenius norms of the weight matrices serve as regularization terms.

Postulate 1: What is the meaning of the superscript "u" in L^u. "unnormalized"?

Postulate 2: It is stated that the domain of N=Theta'Theta can be "reasonably approximated". This seems too vague.

l194: It is stated that Theta has generally full row rank. If this is a necessary assumption, it needs to be made explicit.

Equation (7): I believe that the Phi in the leftmost term should be a Phi' (transposed). Please check.

Section 3.2: I found it very difficult to follow the reasoning in this section. In my opinion it would be useful to show more clearly why the proposed penalty function corresponds to a step-transfer-function

l396: The statement "It is well known that Parity is easiliy computable" should be backed by a reference.
Summary: The idea to use normalized kernels to cast deep learning models as optimization problems with nice convexity properties is nontrivial and innovative. The authors have made an effort to present the material well, but I feel that the contributions of the paper are somewhat overstated as the new approach involves several approximation steps. Also, the claim that block coordinate descent necessarily results in a stationary point seems to be false in general.
Author Feedback
Author rebuttal: Thanks for the thorough reviews. We can address almost all main concerns within the limited space available.

**Convexity of (8):

The objective is clearly convex. To see why the constraint set is also convex, recall from Line 144 that \RR Phi consists of matrices in \rowspan(Phi). Recall also that M and K are PSD, which implies the constraint S\in M\RR K involves the intersection of two convex constraints. To prove this formally, let V={(M, S, K): M and K are PSD, S\in M\RR K}, V1={(M, S, K): M and K are PSD, S\in\colspan(M)}, and V2={(M, S, K): M and K are PSD, S\in\rowspan(K)}. First, note that the set V1 is convex since S1\in\colspan(M1) and S2\in\colspan(M2) implies (S1+S2)\in\colspan(M1+M2) whenever both M1 and M2 are PSD. (This is easy to see since if M1 and M2 are PSD we have \colspan(M1)\union\colspan(M2))\subseteq\colspan(M1+M2), proved by considering the contrapositive: if x'(M1+M2)x=0 hence x'M1x+x'M2x=0, then since x'M1x>=0 and x'M2x>=0 it must follow that x'M1x=0 and x'M2x=0.) V2 is convex for isomorphic reasons. It remains only to show V = V1\intersect V2, since an intersection of convex sets is convex. The \subseteq relationship is straightforward. To show \supseteq, let S = M*P = Q*K for some P and Q; then note that S*K^+*K = S = M*M^+*S, hence M*M^+*S*K^+*K = S.

Therefore (8) is convex. We stand by the assertion that the paper provides a convex formulation of multiple nonlinear layers.

**Quality of the convex relaxation:

The main relaxation introduced in Section 3.1 is replacing the eigenvalue constraint \lambda_i\in{0,1} (implied by M^2=M for PSD M) with 1\geq lambda_i\geq 0 (implied by I\succeq M\succeq 0 for PSD M). Such a relaxation retains sufficient structure to allow, e.g., a 2-approximation of optimal clustering to be preserved even by only imposing spectral constraints [30]. Relaxing Booleanness is not necessarily as consequential as suggested. Although we do not yet have an approximation bound for the full model, experimental results demonstrate that nesting preserves sufficient structure, even with relaxation, to capture relationships that cannot be recovered by shallower architectures.

**Relaxed formulation preserves the essence of deep networks?

To gain further evidence of whether the relaxation preserves any capabilities of a multi-layer architecture, we ran additional comparisons against other baseline methods, as requested.

SVM1: 1-layer SVM
TSVM1a: 1-layer transductive SVM (Sindhwani, Keerthi, SIGIR-2006)
TSVM1b: 1-layer transductive SVM (Joachims, ICML-1999)
NNET2: 2-layer sigmoid neural network, hidden layer size chosen by cross-val
LOC3: 3-layer model, exact (unrelaxed) with local optimization

The same protocol was followed for selecting the parameters of each method and gathering test errors as in the paper. Average test errors obtained:

200/200 CIFAR MNIST USPS COIL Letter
SVM1__ 32.3 12.3 10.3 14.7 4.8
TSVM1a 32.0 10.7 10.3 13.7 3.8
TSVM1b 26.0 10.0 11.0 18.9 4.0
NNET2_ 30.7 11.3 11.2 14.5 4.3
LOC3__ 28.2 12.7 8.0 12.3 7.3

Comparing these results to those for CVX2 and CVX3 shows some important points. First, the baseline methods are not competitive in any case, while CVX3 obtains the best result on each data set. The gap between CVX3 and LOC3 shows that local minima can be a problem in the architecture and data sets considered. The gap between NNET2 and CVX3 (and between CVX2 and CVX3) demonstrates that additional structure is being exploited by the 3-layer model that is not being captured by the 2-layer methods.

**Layer-wise losses and transfers:

Although deep learning research has traditionally focused on exponential family models (e.g. auto-encoders, PFNs and RBMs), these are not the only possibility; continuous latent variable models such as the (particularly successful) rectified linear model and the exponential family harmonium have also been considered. In fact, rectified linear layers are even easier to accommodate in the proposed framework than the current large margin model. Nevertheless, a large-margin approach offers additional sparsity and simplifications; it remains sufficient to prove our main point.

The large margin layer loss (13) does not correspond to a linear transfer between layers, even in terms of the propensity matrix S or normalized output kernel M. As in all large margin methods, the initial loss (12) is a convex upper bound for an underlying discrete loss defined with respect to a step transfer.

**Comparison to the 2-layer model in [26]:

The 2-layer model in [26] employs the objective in Line 178 with the 3rd and 4th terms removed. Our approach differs by using value regularization ||\Theta’ U||^2 instead of ||U||^2. A new postulate 3 is introduced to enforce convexity of the loss for normalized kernels. The key obstacle in extending [26] is that the objective considered (line 178) is not jointly convex.

**Correctness of Algorithm 1:

It is well known that block coordinate descent (BCD) can fail on nonsmooth objectives (such as the one demonstrated by Reviewer_41's example). However, our implementation considered a smooth objective R, as noted in Line 348 (Step 1). (In particular, the max in (13) was smoothed by a softmax, which unfortunately was not made sufficiently clear.) Fortunately, BCD can be correctly implemented for smooth objectives: by Prop 2.7.1 of "Nonlinear Programming" (Bertsekas, 1999), BCD produces a (sub)sequence that converges to a critical point, as long as each block optimization has a unique optimal solution. To ensure uniqueness, we simply used a method equivalent to the proximal method in Proposition 7 of "On the convergence of the block nonlinear Gauss-Seidel method under convex constraints" (L. Grippoa and M. Sciandrone, Operations Research Letters, 26, 127-136, 2000).

We maintain that the optimization algorithm converges to critical points (hence global minimizers, by convexity) for a smooth objective.